# The development of narrative skills in Turkish-speaking children: A complexity approach

Hale Ögel Balaban [1], Annette Hohenberger [2]¤ *

1 Department of Psychology, Bahçeşehir University, Istanbul, Turkey, 2 Department of Cognitive Science, Middle East Technical University, Ankara, Turkey

¤ Current address: Institute of Cognitive Science, University of Osnabrück, Osnabrück, Germany
* annette.hohenberger@uni-osnabrueck.de

**Citation:** Ögel Balaban H, Hohenberger A (2020) The development of narrative skills in Turkish-speaking children: A complexity approach. PLoS ONE 15(5): e0232579. https://doi.org/10.1371/journal.pone.0232579

**Data Availability Statement:** The data underlying the results presented in the study are available from the Center for Open Science (COS) at https://osf.io/vxqpc/.

## Abstract

The present study examines the development of plot, evaluative and syntactic complexity in children's narratives and its relationship with gender, ToM, executive function and linguistic recursive ability. One hundred and five Turkish-speaking children distributed across 4 age groups (four-, five-, seven-eight-, and ten-eleven-year-olds) and 15 adults participated in (a) Elicitation of Narratives Task, (b) Emotional Stroop Task, (c) First- or Second-Order ToM Task (depending on their age), (d) Real-Apparent Emotion Task (four-year-olds), and (e) Comprehension of Complement Clauses Task. Among the three domains of complexity, only plot complexity was found to be related to gender and to develop significantly, in particular at 5 and 7 years of age. Evaluative complexity was low in children in all age groups and was not predicted by any factor. Syntactic complexity was predicted by executive function. These findings are discussed considering the cognitive, linguistic and sociocultural nature of narration.

## Introduction

Narrative is a type of discourse referring to goal-directed events that are sequenced in a causal and temporal order [1]. According to Labov and Waletzky [2], narrative has two main functions: referential and evaluative. Its referential function is to express the events in sequenced clauses that reflect their temporal order. Its evaluative function refers to the expression of the narrator's interpretation of and attitude towards the referential components. Bruner [3] identified corresponding levels of organization of narrative, namely the landscape of action and the landscape of consciousness, respectively. Considering the functions of narrative and its organization, narrative is a complex form of discourse. Creating it requires 'a joint process of event comprehension and language production' [4, p.87], and understanding and explaining behaviors and emotions of others through perspective taking. In the present study, we discern three different notions of complexity in narration: plot, evaluative, and syntactic complexity. We examine how these three domains of complexity develop together over the preschool and school years. Although previous research has covered different domains together and examined their relationship to some cognitive and social factors as discussed below, it is timely to combine all three domains and to unravel their cognitive and linguistic bases by relating

**Funding:** AH and HÖB received a university-internal grant, BAP 07-04-2012-003, from Middle East Technical University (METU), Turkey. The grant amounted to 6.800 Turkish Liras (TL). The funding website is: https://bap.metu.edu.tr/ The funders had no role in study design, data collection and analysis, decision to publish, or preparation of the manuscript.

**Competing interests:** The authors have declared that no competing interests exist.

narrative development to the development of theory of mind (ToM), executive function, and linguistic recursive ability. A further novelty of our study is the choice of Turkish which is typologically different than the mostly studied Indo-European languages such as English.

## Plot complexity

Plot is defined as a sequence of events connected to each other to comprise a meaningful whole [3]. It is the result of the narrator's integration of a setting with thematically, temporally and causally related episodes of events [4, 5]. An analysis of plot complexity is relevant for the referential function of narrative corresponding to the landscape of action.

The plot line includes three main components: 1. the onset referring to a starting event, 2. the unfolding referring to the extension of the events in the story, and 3. the resolution referring to the reaching of an outcome [5]. These components constitute a hierarchical structure with a superordinate goal motivating subordinate goals that are encoded with their purposes and outcomes [4]. Plot complexity is thus revealed in the extent to which the narrator realizes this hierarchical structure in his/her plot line. Developmental increases in plot complexity were found in three- to nine-year-old English-, German-, Spanish-, Hebrew-, Turkish- [5], and Finnish-speaking [6] children, as well as English-speaking Latino and African American children [7, 8]. The preschool and school years seem to be an important developmental period for increases in plot complexity.

## Evaluative complexity

During narration, the narrator sometimes departs from the plot and incorporates his/her interpretation into the narrative, as part of the landscape of consciousness [9]. S/he reports the mental states of the characters, describes the reasons or outcomes of the events and the behaviors of the story characters, or integrates his/her own viewpoint into the narrative. Evaluative complexity results from this perspective taking. Both meta-cognitive and meta-linguistic abilities are required from children in order to conceive and syntactically express such evaluation.

As in the study of Bamberg and Damrad-Frye [9] on English-speaking children, Berman and Slobin [5] found an increase in the use of evaluative devices in narratives of Hebrew-speaking children. On the other hand, Küntay and Nakamura [10] reported no developmental change in the use of evaluative devices by four- to nine-year-old Turkish- and Japanese-speaking children and adults. The comparison of Küntay and Nakamura's findings with those of Bamberg and Damrad-Frye imply possible cross-linguistic and cross-cultural differences in the preferences for evaluative devices. For instance, the use of character speech was less frequent in English than in Japanese narratives whereas causative expressions and hedges were more frequent in English than in Turkish and Japanese narratives. In the light of these contradictory cross-linguistic findings, our study aims to elucidate in particular the development of Turkish children's evaluative abilities.

## Syntactic complexity

Syntactic complexity is a fundamental property of human language and is achieved through the recursive, hierarchic organization of components governed by syntactic principles [11]. The organization of narrative is reflected through syntactic complexity insofar as complex syntactic structures are means to express the coherent causal, temporal and logical order of the reported events. Thus, the domain of syntactic complexity is related to both plot and evaluative complexity.

In narrative literature, syntactic complexity is operationalized in different ways such as the mean length of sentences in words or morphemes [6, 7, 12] and the ratio of the number of

syntactically complex clauses over the total number of clauses [13, 14]. We adopted the latter definition with a focus on subordination. Subordination is one way to create recursive hierarchies in language. In subordination, the constituent on the lower level depends on the one on the higher level. Children acquire complex clauses with subordination during the period of 2 to 4 years of age [15, 16]. Several studies have shown a developmental increase in the use of subordination in the narratives of English-speaking children in the age range of 4 to 12 years [14, 17]. However, Turkish is typologically different than English. In English, syntactically complex structures are mostly finite and formed with subordinating conjunctions. On the other hand, Turkish is an agglutinating language in which syntactically complex structures are mostly nonfinite and usually formed with subordinating suffixes attached to the verb stem [18]. Moreover, the subordinating suffixes are modified depending on the verb stem in accordance with the vowel harmonization rules of Turkish and in some constructions they are followed by the possessive suffix for person, number and case [18]. Considering these cross-linguistic differences and the lack of studies including syntactic complexity in Turkish narratives, our study aims to elucidate the development of syntactic complexity in Turkish-speaking children in relation to their narrative development.

## Multidimensional analysis of narratives

In recent years, developmental studies focused on more than one dimension of complexity: evaluative and plot complexity [19] or evaluative and syntactic complexity [20]. However, even these studies are limited in some respects. First of all, the measures related to plot, syntax and evaluation were only included as indicators of different linguistic skills such as general language abilities and pragmatic skills [19, 20]. Moreover, none of these studies cover wide age ranges although patterns might change over different developmental periods. In the light of these shortcomings, the aim of the present paper is four-fold: First, to study all three domains of complexity together; second, to cover a wide age range comprising the preschool period (fourth and fifth years of life) in which many developmental changes occur in narrative, cognitive and linguistic abilities relevant for the present study [6, 21, 22, 23], middle childhood (seventh and eighth years of life) and late childhood (tenth and eleventh years of life) between which developmental changes in linguistic and socio-cognitive abilities continue to occur [24, 25, 26]; third, to contribute to the cross-linguistic and cross-cultural perspective on narrative development by studying Turkish-speaking children; and fourth, to analyze how the development in each domain of complexity is related to their presumed underlying cognitive, social and linguistic processes.

## Social, cognitive and linguistic processes underlying narrative development

The relationship between social cognition, especially ToM, and narrative competence has been the focus of interest for a long time [20, 27]. First-order ToM abilities–distinguishing between representation and reality, and understanding false belief–develop between 3 and 5 years of life [28, 29]. At around the same time, children's focus shifts from the plot elements to the thoughts, beliefs and intentions of the story characters in their narratives. Due to this coincidence, Astington [30] proposed a relationship between ToM and the use of evaluative devices. Supporting this proposal, Pelletier and Astington [27] found that among four- and five-year-old English-speaking children those who were more successful on ToM tasks could coordinate the actions and the thoughts of the story characters in their narratives better compared to other children who were less successful. Furthermore, the impairment of ToM in autistic children was demonstrated to be related to their difficulties in referring to mental states of the story characters [31, 32]. The relation between evaluative complexity and ToM has also been

studied on the level of second-order ToM. Developmental studies indicated that second-order ToM reasoning develops around age 5 or 6 [22, 23, 33]. Findings on ToM and its relation to evaluative devices were contradictory, though. Fernández [20] showed that second-order ToM abilities of four- and eight-year-old Spanish-speaking children were good predictors of evaluative complexity. On the contrary, second-order ToM abilities were not found to predict the use of mental state terms in children between six and twelve years of age speaking various languages [32, 34, 35]. Besides methodological differences, the diverse outcomes of these studies may be due to the change in the developmental relationship between ToM and evaluative complexity between 4 and 12 years of age [20], i.e., evaluative complexity and second-order ToM may be related at younger ages but not at older ages.

The development of executive function has also been claimed to be related to narrative development considering the parallelism between these two domains as well as their rapid conjoint development during the preschool period [36]. Three cognitive components of executive function are required for the construction of the plot line in a narrative [37]. Shifting, referring to the ability to revise responses flexibly according to changing tasks or mental sets is necessary 'to recall and integrate content for the story narrative' [37, p. 827]. Updating, responsible for keeping and monitoring representations in memory and modifying them according to new information is required to '[recall] prior episodes or episodic components in order to appropriately elaborate the story' [37, p.827]. Inhibition refers to the ability to selectively attend to specific tasks, to complete the task without being influenced by distractors and to inhibit prepotent responses [38]. It is essential for narration, because the narrator needs to focus on the narrative and complete it without getting distracted. In addition, the narrator has to inhibit extraneous or inappropriate comments for a coherent narrative [37, 38]. Executive function is also needed for the formation of syntactic structures in narrative. Forming syntactically complex clauses requires planning, holding the grammatical units in working memory, inhibiting irrelevant information, binding the units together and forming clauses successively. All of these processes can be attributed to the inhibition component of executive function. Supporting this claim, in a six-month cross-lagged study, Friend and Bates [36] found that 'the ability to focus attention and resist distraction at 4.5 years confers benefits in the ability to construct a complex and coherent narrative at 5 years of age' (p. 10). However, there was no convergent relationship between executive and narrative skills at either 4.5 or 5 years of age. The study by Drijbooms et al. [38] demonstrated a relationship between inhibition and syntactic complexity in written narratives in middle childhood. These findings suggest that the relationship between executive function and narration might change throughout development.

The development of language as a representational system might be another contributor to narrative development. In Nelson's view [39], language is a system that is used to represent one's own representations for oneself, reflect these representations to others, hold others' representations and represent others' representations as different from own representations. To the extent that self-representation is crucial for the landscape of consciousness and is typically expressed through complex syntax, the present study explores whether recursive syntactic abilities are related to evaluative complexity. Finally, considering recursive syntactic abilities as an indicator of general syntactic abilities, we claimed that they are also related to the development of syntactic and plot complexity in narrative. Children with higher syntactic abilities might form more syntactically complex clauses in their narrative. They also might use less cognitive resources to form these structures and have more cognitive resources to construct the plot of the narrative.

In summary, the present study takes a complexity approach and brings together different domains of complexity in narratives, namely plot, evaluative and syntactic complexity, with cognitive, social and linguistic processes which might be underlying the development in these

domains. We hypothesize that the measures of plot complexity, evaluative complexity and syntactic complexity increase with age. In addition, ToM, executive function and linguistic recursive ability are expected to increase with age. First-order ToM abilities of the youngest children and second-order ToM abilities of older children are expected to predict evaluative complexity in narrative. In terms of executive function, inhibitory control is expected to predict plot complexity and syntactic complexity. Finally, linguistic recursive ability is expected to predict plot, evaluative and syntactic complexity (for a graphical depiction of our hypotheses, see S1 Fig).

Child gender is considered to play a role in how parents and children narrate about past events or in shared book-reading activities [40, 41]. However, it is mostly ignored in studies on fictitious narratives [5, 6, 7, 37]. Here, we take child gender into account and explore whether there are gender differences in the development of narrative complexity.

## Method

### Participants

A total of one-hundred-five children in 4 age groups and fifteen adults (as a reference group) participated in this cross-sectional study. Table 1 presents the distribution of the participants according to age groups and gender.

All of the participants were native Turkish speakers. Four- and five-year-olds were recruited from four kindergartens in Istanbul. Seven-, eight-, ten- and eleven-year-olds were recruited from three primary and secondary public schools in Istanbul. These kindergartens and schools were located in districts with population of mostly middle socioeconomic status.

All of the child participants were reported to be normally developing by their teachers. In return for their participation, four- and five-year-olds received a sticker and older child participants received a pencil. Adults were recruited from Istanbul Bilgi University. They were undergraduate students in the Psychology department. In return for their participation, they received two credits in the Experimental Psychology or the Cognitive Psychology courses. The study was approved by the Ethics Committee of Middle East Technical University, Ankara. Consent was taken from the Directorate of National Education in Istanbul, the school principals, the adult participants and the parents of the child participants.

### Materials

For each task and each domain of complexity, inter-rater reliability analyses based on the consensus estimates method indicating the agreement between the raters were conducted on 25% of data. A psychology graduate student in Bahçeşehir University in Istanbul was trained for transcribing and coding data of randomly selected 30 participants. Differences were resolved through discussion between the experimenter and the second rater.

**Table 1. Descriptive statistics (gender, age) of participants according to age groups.**

| Age group | No of participants | | Age | |
| --- | --- | --- | --- | --- |
| | Male | Female | M | Range |
| 4 | 11 | 7 | 4.4 | 3.9–4.11 |
| 5 | 11 | 11 | 5.4 | 5.0–5.10 |
| 7 & 8 | 12 | 21 | 7.9 | 7.0–8.9 |
| 10 & 11 | 16 | 16 | 11.3 | 10.4–11.11 |
| Adults | 2 | 13 | 21.2 | 20.3–23.2 |

**Elicitation of narratives.** To elicit the narratives, Mayer's 24-page wordless picture book 'Frog, where are you?' [42] was used. This book was suitable for the aims of the present study, because it has a strong plot-line and it invites emotional and cognitive appraisal of the various situations the protagonist encounters. Moreover, it has been used in numerous other studies about narrative development [5, 10, 36, 43]. First, the experimenter introduced the book to the participants by stating that the book was a wordless picture book depicting a story about a boy, a dog and a frog. She asked them to look at all pages of the book in the presented order, and then to tell the story in their own words while looking at the pictures on the pages. She emphasized that the story should include the experiences, the emotions and the thoughts of the story characters. The experimenter answered only the participants' questions about the identity of the story characters (e.g. the deer, the gopher). Aside from this, she did not interfere with the narration. After the participants had finished their stories, the experimenter thanked them and introduced the next task.

**Theory of mind (ToM) tasks.** *First-order ToM Tasks*. *Change of Location Task*. The change of location task developed by Wimmer and Perner [29] was used to assess ToM abilities of four-year-old participants. Similar to the original version of the false belief task, a story was acted out with toys in which the participants had to infer a character's belief about the (re-)location of an object. If participants correctly answered all of the memory control questions and the false-belief question, they passed the task and obtained 1 point. The inter-rater reliability was found to be 100% ($K = 1.00$, $p < .05$).

*Real-apparent emotion task*. Understanding of others' emotions is relevant to the evaluative abilities in narrative. Thus, the real-apparent emotion task included in Wellman and Liu's [44] ToM scale was used to assess four-year-old participants' ability to differentiate between the emotion that a person feels and the emotion s/he displays. The experimenter told the participants a story about a girl at the end of which the participants were asked to report how the girl would really feel and what the facial expression of the girl would be by pointing to cartoon faces with sad, happy and neutral expressions. If the participants responded to the real emotion question by pointing to the sad cartoon face and to the apparent emotion question by pointing to the happy or neutral cartoon face, they passed the task and obtained 1 point. Any other response combination was evaluated as incorrect. The inter-rater reliability was found to be 95.45% ($K = .82$, $p < .01$).

An overall first-order ToM score was computed by adding the change of location task score and the real-apparent emotion task score for the four-year-old participants. The first-order ToM score ranged between 0 and 2.

*Second-order ToM tasks*. To assess ToM abilities of five- to eleven-year-old and adult participants, the experimenter told two stories in which the participants had to infer a character's belief about another character's belief. During the story telling, drawings created by Flobbe [45] to depict the stories were presented to foster comprehension. The first story, the chocolate bar story, was adapted from Perner and Wimmer's [22] set of second-order ToM stories. In the story, a boy puts a chocolate bar in a drawer. In his absence, his sister moves the chocolate bar to a toy-box. Through the window, the boy sees that his sister relocates the bar; however, she is not aware of that. At this point, the experimenter asked the reality control question ('Where is the chocolate bar now?'), the 1st order ignorance question ('Does the boy know that his sister moves the chocolate to the toy-box?'), and the linguistic control question ('Does his sister know that the boy sees her relocating the chocolate?'). Later, the boy comes back and states that he wants to eat his chocolate. The experimenter presented the 2nd order false-belief question ('Where does his sister think that the boy will look for the chocolate?'). The second story, the birthday present story, was adapted from Sullivan et al.'s [23] set of stories to Turkish by Arslan et al. [33]. Participants' responses were evaluated separately for each story. If the

participants correctly answered the 2<sup>nd</sup> order false-belief and control questions regarding one story, they passed that story and obtained 1 point. An overall second-order ToM score was computed by adding the points from the two stories. It ranged between 0 and 2. The inter-rater reliability was found to be 97.50% in the chocolate bar story ($K = .91$, $p < .001$) and 96.67% in the birthday present story ($K = 1.00$, $p < .001$).

*Executive-Function Task—Emotional Stroop Task.* The Emotional Stroop Task developed by Lagattuta, Sayfan and Monsour [46] was used to assess executive function. It has been demonstrated to be a measure of inhibitory control that can capture individual differences over an age range starting from the 4th year of life and expanding through adulthood [46], thus in order to avoid ceiling effects for the older children, we chose this task. Like other inhibition tasks, it puts demands on working memory [36, 47], because it requires keeping (the rule) how to respond to the stimuli in mind during the task. Moreover, the task has an emotional component which taps the emotional aspects of narrative ability.

Before the administration of the task, the experimenter checked whether the participants could identify happy and sad faces. Then, she introduced the rule of the task by explaining that they would play an "opposite game": when she showed the happy face, the participants had to respond saying *üzgün* 'sad' and when she showed the sad face, the participants had to respond saying *mutlu* 'happy'. After four training trials, they continued with testing trials in which 10 happy and 10 sad faces were randomly presented. No positive or negative feedback was given. However, if the participants made 4 errors in a row, the experimenter reminded them of the rule.

The ratio of the number of correct responses to the maximum number of possible correct responses (20) was computed as the executive function score. The inter-rater reliability was found to be 96.50% ($K = .91$, $p < .001$).

*Linguistic recursivity task—Comprehension of complement clauses task.* Altan [48] had developed a task to assess children's ability to comprehend complement clauses inspired by a task developed by Thornton [49]. Her task was adapted for the present study to assess linguistic recursive ability. At the beginning of the task, the experimenter introduced a turtle puppet to the participants and informed them that they would ask questions presented by the experimenter to this puppet. The experimenter presented six single-embedded clauses (e.g., *Kaplumbağaya dün akşam televizyonda ne izlediğini sorar mısın*? 'Could you ask the turtle what he watched yesterday evening on TV?'), and six double-embedded clauses (e.g., *Kaplumbağaya dün ne yaptığını sana anlatmasını söyler misin*? 'Could you ask the turtle to tell you what he did yesterday?') to the participants one by one in 12 trials. Participants were expected to comprehend these syntactically complex structures and direct them to the turtle puppet after rephrasing them into a question format. The order of these clauses was determined randomly and was the same for all participants.

Participants' correct responses to the clauses with single embedding were scored as 1. Their correct responses on the clauses with double embeddings were scored as 2. Incorrect and unrelated responses were scored as 0. If participants could understand the complement clauses with double embeddings, but could reproduce only some part of the question (e.g., *Dün ne yaptın*? 'What did you do yesterday?' in response to *Kaplumbağaya dün ne yaptığını sana anlatmasını söyler misin*? 'Could you ask the turtle to tell you what he did yesterday?'), their responses were scored as 1. Finally, if they divided the double-embedded clauses into two separate clauses (e.g., *Dün ne yaptın*? *Bana anlatır mısın*? 'What did you do yesterday? Can you tell me?'), their responses were scored as 1.5. The maximum possible score on this task was 18. The ratio of the total scores of participants to the maximum possible total score was computed as the linguistic recursivity score. The inter-rater reliability was found to be 93.61% ($K = .87$, $p < .001$).

## Procedure

Data from the child participants were collected in a silent room in their kindergartens and schools. Data from the adult participants were collected in a silent classroom in Istanbul Bilgi University. All tasks were administered individually to all participants by the same native Turkish-speaking experimenter. To four-year-old participants, the tasks were presented in two sessions. There was a 4 to 7 days long interval between the sessions. In the first session, the order of the tasks was: (1) Elicitation of the Narrative Task and (2) Emotional Stroop Task. In the second session, the order of the tasks was: (1) Change of location Task, (2) Real-apparent Emotion Task, (3) Comprehension of Complement Clauses Task. Each session lasted approximately 10 minutes. To participants in other age groups, the tasks were administered in a single session in the fixed order of: (1) Elicitation of Narrative Task, (2) Emotional Stroop Task, (3) Second-order ToM Task, (4) Comprehension of Complement Clauses Task. This session lasted 15–20 minutes. All of the tasks were video-recorded for transcription and coding. The video-recorder on top of a tripod was put in front of the participants at a distance without distracting them.

## Transcription

Video-recordings of the narratives were transcribed by the experimenter using EUDICO Linguistic Annotator (ELAN) (http://tla.mpi.nl/tools/tla-tools/elan/) developed at the Max Planck Institute for Psycholinguistics, Nijmegen, Netherlands [50].

## Coding criteria for the domains of complexity

**Plot complexity.**  Plot complexity was coded according to the criteria constructed by Ayas-Koksal [51] based on the plot components suggested by Berman and Slobin [5] for the book 'Frog, where are you?' [42]. There were 4 main components with their subcomponents: a) the onset including the introduction of the characters, the setting and the disappearance of the frog as the main event; b) the unfolding including the experiences of the boy and the dog with different animals while searching for the frog; c) the search theme including the references to the searching of the frog; d) the resolution including the boy's finding of the frog. The manual for coding and scoring is presented in S1 Table. The plot complexity score was computed by adding up all the subcomponent scores and calculating the ratio of this total score to the maximum possible score (19 points). The inter-rater reliability was found to be 86.88%.

**Evaluative complexity.**  All clauses were coded as either referential or evaluative. A clause expressing a scene, an event or information directly observable in the pictures of the book was coded as referential, including perceptual states of the protagonist (e.g., seeing, hearing, smelling). A clause including an evaluation of the narrator regarding the events or the story characters, or stating the point of view of the narrator him/herself was coded as evaluative. Evaluative clauses had at least one item from the evaluative device categories presented in S2 Table. They were adapted and modified from Küntay and Nakamura [10]. Similar categories were also reported in studies about evaluation in narrative [20, 52].

An evaluative complexity score was created by taking the ratio of the number of evaluative clauses over the total number of clauses (the sum of the number of referential and evaluative clauses). The inter-rater reliability was found to be 73.84% ($K = .84$, $p < .001$).

**Syntactic complexity.**  Each communication unit (C-unit) defined as a main clause with its associated subordinate clauses was coded as either simple or complex. In Turkish, subordinate clauses come in three kinds, as noun phrases, adverbial phrases and relative clauses [18]. Göksel and Kerslake [18] summarized their marking as in S3 Table. The subordinate clauses in narratives were identified and classified accordingly. A syntactic complexity score was

calculated as the ratio of the number of complex C-units (i.e. C-units with at least one subordinate clause) to the total number of C-units. The inter-rater reliability was found to be 77.84% ($K$ = .88, $p <$ .001).

## Results

Data from adult participants were excluded from the statistical analyses in order not to distort the results of the children, but included in the figures or tables for only descriptive purposes so that the child participants' performance can be compared to the performance of the adult participants. All hypotheses were tested in two-tailed tests. To analyze participants' performance on each task, the effect of age and the effect of gender were examined together. Each age and gender group's ToM, executive function and linguistic recursivity scores; and the plot, evaluative and syntactic complexity scores are presented in Table 2.

### ToM, executive function and linguistic recursivity

Two-way ANOVAs with age and gender as the independent variables showed a significant effect of age on ToM, executive function and linguistic recursivity scores, $F$ (3, 95) = 10.43, $p$ = .000, $\eta_p^2$ = .25, observed power = 1.00; $F$ (3, 97) = 9.74, $p$ = .000, $\eta_p^2$ = .23, observed power = 1.00; and $F$ (3, 92) = 13.15, $p$ = .000, $\eta_p^2$ = .30, observed power = 1.00 respectively. The effect of gender and the interaction between age and gender on each measure were not found to be significant. Results of the repeated planned contrasts indicated that 5-year-olds' ToM score was lower than that of 7- and 8-year-olds which was lower than that of 10- and 11-year-olds. The linguistic recursivity score of 3- and 4-year-olds were lower than that of 5-year-olds which were lower than 7- and 8-year-olds' score. Five-year-olds' executive function score was lower than 7- and 8-year-olds' score.

### Domains of complexity

A (4) age group x (2) gender MANOVA with age and gender as the between-subject independent variables and plot complexity, evaluative complexity and syntactic complexity scores as the dependent variables was conducted. There was a significant effect of age on the domains of complexity, $V$ = 0.51, $F$ (9, 288) = 6.61, $p$ = .000, $\eta_p^2$ = .17. The effect of gender was also

**Table 2. Means (standard deviations) of the ToM, executive function and linguistic recursivity scores and the plot, evaluative and syntactic complexity scores by age.**

| Tasks | Age groups | | | | | | | | | | | | |
|---|---|---|---|---|---|---|---|---|---|---|---|---|---|
| | 4 | | | 5 | | | 7 & 8 | | | 10 & 11 | | | adult |
| | Male | Female | Total | Male | Female | Total | Male | Female | Total | Male | Female | Total | Total |
| ToM score | 0.30 (0.48) | 0.57 (0.79) | 0.41 (0.62) | 0.27 (0.65) | 0.73 (0.79) | 0.50 (0.74) | 1.00 (0.74) | 1.00 (0.84) | 1.00 (0.79) | 1.38 (0.62) | 1.47 (0.64) | 1.42 (0.62) | 1.87 (0.35) |
| Executive Function score | 0.71 (0.27) | 0.71 (0.24) | 0.71 (0.25) | 0.75 (0.14) | 0.82 (0.11) | 0.79 (0.13) | 0.88 (0.08) | 0.89 (0.14) | 0.89 (0.12) | 0.91 (0.07) | 0.92 (0.07) | 0.92 (0.07) | 0.98 (0.04) |
| Linguistic recursivity score | 0.56 (0.16) | 0.55 (0.27) | 0.55 (0.20) | 0.58 (0.17) | 0.73 (0.15) | 0.66 (0.18) | 0.75 (0.13) | 0.75 (0.11) | 0.75 (0.12) | 0.82 (0.07) | 0.78 (0.98) | 0.80 (0.08) | 0.90 (0.09) |
| Narrative | | | | | | | | | | | | | |
| Plot complexity | 0.39 (0.21) | 0.53 (0.17) | 0.44 (0.20) | 0.53 (0.115) | 0.69 (0.13) | 0.61 (0.16) | 0.73 (0.11) | 0.76 (0.09) | 0.75 (0.10) | 0.73 (0.10) | 0.76 (0.14) | 0.75 (0.12) | 0.87 (0.08) |
| Evaluative complexity | 0.22 (0.12) | 0.39 (0.25) | 0.28 (0.18) | 0.31 (0.11) | 0.26 (0.10) | 0.29 (0.11) | 0.24 (0.11) | 0.24 (0.11) | 0.24 (0.10) | 0.25 (0.09) | 0.28 (0.10) | 0.26 (0.10) | 0.40 (0.08) |
| Syntactic complexity | 0.11 (0.09) | 0.27 (0.20) | 0.17 (0.15) | 0.19 (0.14) | 0.20 (0.11) | 0.20 (0.12) | 0.17 (0.10) | 0.18 (0.12) | 0.18 (0.11) | 0.26 (0.09) | 0.23 (0.09) | 0.24 (0.09) | 0.44 (0.11) |

significant, $V = 0.12$, $F (3, 94) = 4.15$, $p = .008$, $\eta_p^2 = .12$. There was a significant interaction effect between age and gender, $V = 0.18$, $F (9, 288) = 2.09$, $p = .03$, $\eta_p^2 = .06$.

Following the MANOVA, separate (4) age group x (2) gender two-way ANOVAs were conducted on the three dependent variables.

**Plot complexity.** On the plot complexity score, the effect of age was significant, $F (3,96) = 22.07$, $p = .000$, $r = .11$, $\eta_p^2 = .41$, observed power = 1.00. Planned repeated contrasts demonstrated that four-year-old participants' plot complexity score was lower than that of five-year-old participants which was lower than that of seven- and eight-year-old participants. The effect of gender was significant, $F (1, 96) = 11.21$, $p = .001$, $r = .02$, $\eta_p^2 = .11$, observed power = .91. Female participants ($M = 0.72$, $SD = 0.14$) had higher plot complexity scores than male participants ($M = 0.61$, $SD = 0.20$). The interaction effect between age and gender was not significant, $F (3,96) = 1.77$, $p = .16$.

**Evaluative complexity.** On the evaluative complexity score, there were no effect of age and no effect of gender, $F (3,96) = 1.06$, $p = .37$ and $F (1,96) = 2.09$, $p = .15$, respectively. The interaction effect between age and gender was significant, $F (3,96) = 3.08$, $p = .03$, $\omega = .10$, $\eta_p^2 = .09$. However, repeated planned contrasts did not reveal significant gender differences between subsequent age groups.

**Syntactic complexity.** Results revealed no significant effect of age and no significant effect of gender on the syntactic complexity score, $F (3,96) = 2.01$, $p = .12$ and $F (1,96) = 2.47$, $p = .12$, respectively. The interaction effect between age and gender was not significant either, $F (3,96) = 2.32$, $p = .08$.

**Relationship between the domains of complexity and ToM, executive function and linguistic recursive ability.** To analyze the relationships between each domain of complexity and ToM, executive function, and linguistic recursive ability, first separate correlational analyses on each domain of complexity were run. Then, only the predictors that were found to significantly correlate with each domain of complexity were included in the subsequent regression analysis. The variance inflation factor (VIF) value close to 1.00 and the tolerance statistics above 0.2 indicated no multicollinearity [53].

Considering the use of the first-order ToM tasks in four-year-olds and the second-order ToM tasks in older child participants, distinct correlational analyses were computed for four-year-olds and older participants. Results of the correlational analyses for younger and older children are given in Tables 3 and 4, according to our hypotheses.

None of the ToM, executive function and linguistic recursivity scores was correlated with the plot complexity score in four-year-olds; thus, no further regression analysis was conducted.

**Table 3. Bivariate correlations between plot, evaluative and syntactic complexity scores, and ToM, executive function, and the linguistic recursivity scores in four-year-old children.**

|  | 1 | 2 | 3 | 4 | 5 | 6 |
|---|---|---|---|---|---|---|
| 1. Plot | 1 | 0.43 | .29 | .29 | .25 | .28 |
| 2. Evaluative |  | 1 | .91*** | -.02 | .43 | .41 |
| 3. Syntactic |  |  | 1 | .11 | .45 | .34 |
| 4. ToM |  |  |  | 1 | .15 | .01 |
| 5. Executive Function |  |  |  |  | 1 | .51* |
| 6. Linguistic recursivity |  |  |  |  |  | 1 |

*p < .05

** p < .01

***p < .001

**Table 4. Bivariate correlations between plot, evaluative and syntactic complexity scores, and ToM, executive function, and the linguistic recursivity scores in children older than 4-years of age.**

|  | 1 | 2 | 3 | 4 | 5 | 6 |
|---|---|---|---|---|---|---|
| 1. Plot | 1 | 0.03 | .10 | .27* | .32** | .37*** |
| 2. Evaluative |  | 1 | .28** | -.18 | -.01 | -.25* |
| 3. Syntactic |  |  | 1 | .08 | .17 | .04 |
| 4. ToM |  |  |  | 1 | .37*** | .41*** |
| 5. Executive Function |  |  |  |  | 1 | .28** |
| 6. Linguistic recursivity |  |  |  |  |  | 1 |

*p < .05

** p < .01

***p < .001

For older participants, for whom significant correlations between the three predictors and plot complexity had been found, hierarchical regression analysis was conducted with age and gender as the predictors in the first step, and ToM, executive function and linguistic recursivity scores as the predictors in the second step through the enter procedure. In the first step, the model was significant, $F(2, 81) = 13.57$, $p < .001$, $R^2 = .25$, adjusted $R^2 = .23$. As displayed in Table 5, age and gender were significant predictors. In the second step, the model was significant again, $F(5, 78) = 6.60$, $p < .001$, $R^2 = .30$, adjusted $R^2 = .25$ $\Delta R^2 = .05$, p of $\Delta R^2 > .05$. Age and gender were significant predictors.

Because different ToM scores were not found to correlate with the evaluative and syntactic complexity scores of four-years-old and older children, age groups were combined for the following analyses. Table 6 presents the results of the correlational analyses and Table 7 the results of the regression analyses, respectively.

Because none of the ToM, executive function and linguistic recursivity scores was correlated with the evaluative complexity score, no regression analysis was conducted.

On the syntactic complexity score, a hierarchical regression analysis was conducted with age and gender as the predictors in the first step, and the executive function score as the predictor in the second step through the enter procedure. In the first step, the model was not significant, $F(2, 101) = 2.19$, $p = .12$. In the second step, the model became significant, $F(3, 100) = 3.20$, $p = .03$, $R^2 = .09$, adjusted $R^2 = .06$, $\Delta R^2 = .05$, p of $\Delta R^2 = .03$. The executive function score was found to be a significant predictor.

**Table 5. Results of the multiple regression analyses predicting plot complexity in children older than 4 years of age.**

| Plot complexitiy | Unstandardized Coefficients | | Standardized Coefficients | t | p |
|---|---|---|---|---|---|
|  | B | SE | β |  |  |
| 1 (Constant) | .46 | .05 |  | 8.40 | .000 |
| Age | .07 | .02 | .41 | 4.24 | .000 |
| Gender | .08 | .03 | .31 | 3.20 | .002 |
| 2 (Constant) | .28 | .11 |  | 2.50 | .014 |
| Age | .05 | .02 | .28 | 2.41 | .018 |
| Gender | .07 | .03 | .27 | 2.79 | .007 |
| ToM | .00 | .02 | .00 | -0.02 | .982 |
| Executive Function | .12 | .12 | .11 | .98 | .331 |
| Linguistic recursivity | .20 | .11 | .20 | 1.86 | .067 |

**Table 6. Bivariate correlations between evaluative complexity, syntactic complexity scores, and executive function, and the linguistic recursivity score in all children.**

|  | 1 | 2 | 3 | 4 |
|---|---|---|---|---|
| 1. Evaluative | 1 | .47*** | .14 | -.04 |
| 2. Syntactic |  | 1 | .29** | .16 |
| 3. Executive Function |  |  | 1 | .48*** |
| 4. Linguistic recursivity |  |  |  | 1 |

*p < .05

** p < .01

***p < .001

## Discussion

The goal of the present study was to examine the development of plot, evaluative and syntactic complexity and the relationship between these domains and ToM, executive function and linguistic recursive ability that were found to increase within the studied age range.

### Development of domains of complexity and their predictors

**Plot complexity.** Confirming our hypothesis that the domain of plot complexity would increase with age, developmental changes in plot complexity were observed between 4 and 5 years of age, and 5 and 7–8 years of age. These changes suggest that there is a transition around the age of 5 during the preschool years, and another transition around the age of 7 and 8 during the early school years. Similar developmental trends have also been observed in English-, German-, Hebrew-, Spanish-, and Finnish-speaking children [4, 5, 6, 7, 8, 54]. They support Berman and Slobin's [5] suggestion that 'general cognitive and expressive development is responsible, over and above the demands and constraints of acquiring a particular native tongue' (p. 43). Plot complexity of seven and eight-year-old and ten- and eleven-year-old participants did not differ from each other and did not reach the level of the adult participants. These results imply further developmental changes in narrative skills during adolescence toward adulthood as also noted by Berman and Slobin [5]. All these developmental changes in plot complexity leading to more hierarchical structure in narrative, might be related to the growing ability to encode and report the events in a goal-oriented organization, starting around the age of 5 and continuing toward adulthood [4].

In the present study, gender was considered as a factor related to narrative abilities. With regard to plot complexity, girls were found to generate narratives with higher levels of complexity than boys. In the literature, studies on the development of plot structure did not report

**Table 7. Results of the multiple regression analyses predicting syntactic complexity.**

| Syntactic Complexity | Unstandardized Coefficients | | Standardized Coefficients | t | p |
|---|---|---|---|---|---|
|  | B | SE | β |  |  |
| 1 (Constant) | 13.50 | 3.29 |  | 4.10 | .000 |
| Age | 2.06 | 1.08 | .19 | 1.92 | .058 |
| Gender | 1.46 | 2.28 | .06 | 0.64 | .522 |
| 2 (Constant) | 2.11 | 6.01 |  | 0.35 | .726 |
| Age | .81 | 1.19 | .07 | .68 | .500 |
| Gender | .98 | 2.24 | .04 | .44 | .663 |
| Executive function | 17.90 | 7.97 | .25 | 2.25 | .027 |

such a gender difference [20]. In some studies, gender was not even taken into account [5, 6, 7, 37]. However, Nicolopoulou [55] studied 3- to 5-year-old English-speaking children's freely generated narratives and concluded that children have highly distinguished gender-related narrative styles. The plot line of the story depicted in 'Frog, where are you?' includes some parallels with the narrative style of girls reported by Nicolopoulou. Both of them include a home setting as the starting and end points. Characters are related to each other in social relationships. At the end, the disrupted social order is restored. These similarities suggest that the picture book presents the participants a scenario which might be more familiar to girls than boys although the protagonist is a boy. Consequently, compared to boys, girls might have performed better in incorporating the plot components. However, whether data from Nicolopoulou's study can be generalized to older children as in the present study is questionable. Further research examining gender differences in plot complexity in various age groups and in different narrative contexts will provide a better understanding of their pattern and underlying reasons.

Plot complexity was expected to be predicted by executive function and the linguistic recursive ability. Although this expectation was not supported, there was a positive correlation between plot complexity and these other cognitive/linguistic factors in children older than 4 years of age. The positive relationship between plot complexity and linguistic recursive ability, which was approaching significance in the multiple regression analysis, suggested that children who have higher linguistic recursive ability may have higher syntactic competence as a consequence of which they may need less cognitive resources to form the syntactic units during narrative production and have more cognitive resources left to focus on the plot elements. Beyond being a mere resource issue, linguistic recursive ability might overlap with plot complexity in terms of the inclusion of hierarchical representations. A general ability to embed representations into each other to form a hierarchical structure (for the plot as well as for the language) might underlie both of them and their relationship with each other.

The relationship between plot complexity and executive function matches the finding of previous correlational studies focusing on the attention and shifting components of executive function [37, 56]. It suggests that in addition to shifting and attention, inhibition might also be related to plot complexity and in future studies to unravel the predictive effect of executive function shifting, attention and inhibition tasks should be employed together.

**Evaluative complexity.** The domain of evaluative complexity was found not to differ across age groups. This finding is against our hypothesis; however, it is consistent with some previous findings showing no changes in the use of evaluative devices within the studied age range [9, 10, 14, 25]. In all age groups, approximately 27% of all clauses were evaluative clauses. This finding suggests that starting from the age of 4 children have a basic notion of narrative and can integrate the landscape of consciousness reflected by evaluative complexity into their narratives to some extent. However, further developments in evaluative complexity may only be achieved during adolescence and young adulthood, as suggested by the higher results of the adult group.

Evaluative complexity was not found to be predicted by ToM. This finding contradicted previous studies in the literature that had found a predictive effect of ToM on evaluative complexity [20, 27]. On the other hand, they match with other findings showing no relationship between ToM and evaluative complexity [32, 34, 35]. One reason for the lack of this relationship in the four-year-olds can be the observed floor effect in their performance on the first-order ToM tasks. The change of location and the real-apparent emotion tasks are more difficult than other ToM tasks, as Wellman and Liu [44] found with their ToM battery. Another reason might be the gap between ToM competence and its spontaneous use to describe the emotional and mental states of others and to interpret the reason of the events and behaviors [35]. Although adults in the present study performed close to the ceiling level on ToM tasks,

the extent of their use of evaluative devices was limited as well. This further supports the idea that 'having a ToM is different from using one's ToM capacities to describe other people and explain their behavior' [35, p. 193]. This might be especially true in the context of narratives. Narratives are complex tasks with high cognitive and linguistic demands [14]. This load might hinder the use of ToM abilities as a result of which the use of evaluative devices does not reflect the actual mindreading capacities of the narrators [1]. The third reason for the lack of the relationship between ToM and evaluative complexity might be narrators' preference to remain on an implicit level. In a narrative, the actions of the characters carry their intentions. If the narrator presents these hidden intentions explicitly, then the charm of the narrative might get lost. In that sense, narrating is a collaborative action with narrators and listeners co-constructing the landscape of consciousness, but more implicitly than explicitly. This collaboration at the implicit level might be influenced by culture as discussed below.

Evaluative complexity was not found to be predicted by linguistic recursive ability as well. The task used to measure linguistic recursive ability was assessing only the comprehension and reproduction of complement clauses. Among evaluative devices analyzed in the present study, mostly cognitive state verbs such as *san* 'suppose', *düşün* 'think', *anla* 'understand' etc. require the use of complement clauses. The scarcity of the use of these verbs might result in the lack of the relationship between evaluative complexity and linguistic recursive ability. Measuring linguistic recursive ability through the production of other complex syntactic structures is desired in the future.

**Syntactic complexity.** In all age groups, approximately 20% of all C-units formed by the children in their narratives were syntactically complex. This finding does not support the hypothesis that the level of syntactic complexity would increase with age. Several studies showed developmental increase in the use of syntactically complex clauses in narratives of English- [14, 17], Cantonese- [12] and Finnish-speaking [6] children. The present study does not support the findings of these studies. This difference might be caused by typological differences between Turkish and other languages. For instance, in English and Finnish syntactically complex structures are formed mostly with subordinating conjunctions and subordinate clauses are finite [56, 57]. In Turkish, syntactically complex structures are formed mostly with subordinating suffixes attached to the verb stems and subordinate clauses are mostly nonfinite [18]. Moreover, in Turkish the subordinating suffixes are modified depending on the verb stem in accordance with the vowel harmonization rules [18]. The use of some subordinating suffixes such as -mE and -mEk are acquired early, while the use of some others such as -DIK and–(y)AcAK is a late achievement because of their morphosyntactic complexity. Furthermore, Turkish relative clauses were shown to be acquired later than in other languages, again because of their morphosyntactic complexity [58, 59]. Combined with the demands of narration, the complexity of these structures might result in their low extent of use in our Turkish sample. Yet, adult participants were found to use twice as many syntactically complex clauses as child participants. This difference between the child and adult participants suggests developmental changes in the use of syntactically complex clauses in narratives after the age of 11 until adulthood.

The hypothesis that executive function would predict syntactic complexity was supported. This finding extends Drijbooms et al.'s finding [38] in written narratives of fourth grade children to oral narratives of children in an even wider age range. The formation of clauses with embedded subordination requires planning, holding the syntactic units in mind, nesting the subordinate clause under the superordinate one, and inhibiting the completion of the superordinate one until the completion of the subordinate one. This is particularly true for Turkish with its verb-final word order, i.e., the matrix predicate follows the embedded clause. Executive function might play a role in all of these processes.

Executive function was found to account for only 7% of the variation in syntactic complexity suggesting that there are other predictors. Although expected, linguistic recursive ability was not found to be a significant predictor, under a conservative (2-tailed) statistical approach. This lack of the relationship might be attributed to the context and cognitive load of narration. Although children's syntactic competence increased with age as shown by their performance in the Comprehension of the Complement Clauses Task in the present study, this increased ability might not be reflected in narratives due to the high demands of the narrative task. This possibility is also consistent with Slobin's [60] claim that the 'difficulty lies in the packaging of information for narrative purposes'. Moreover, there was not much variance in syntactic complexity scores of children in each age group. This might have influenced the statistical results of the relevant regression analyses and led to the apparent lack of a predictive relationship between syntactic complexity and other factors.

Among the dimensions of complexity, syntactic complexity was found to be related to evaluative complexity. There was a highly positive correlation between them in 4-year-olds and a moderate one in older children. One reason of this relationship may be two simultaneous functions of character speech in narratives. They are one of the evaluative devices and also a means to create syntactically complex clauses. Especially, younger children use them frequently to refer to the internal states of the story characters. In older children, the use of character speech decreases and with it the high correlation between evaluative and syntactic complexity. The second reason might be the correspondence between the use of evaluative devices and the required syntactically complex structures (e.g., cognitive state terms- selecting complement clauses, causal expressions selecting adverbial clauses). Further research focusing on syntactic complexity not only in terms of the use of subordination but also in terms of diverse syntactic structures and its relationship with other domains of complexity might provide a better insight into its development.

Concerning all three domains of complexity in narrative we found that if, they were only related with cognitive abilities in the older but not in the youngest age group. This finding suggests that at four years of age, Turkish children's narrative and their predictive cognitive abilities are not yet developed enough and that only from five years onwards such relations gain predictive power. Moreover, at four years of age, evaluative and syntactic complexity are highly entangled, through the frequent use of character speech. Thus, our study delineates the lower age limit at which predictive relations between narrative and cognitive abilities start developing–but not the upper age limit since evaluative and syntactic complexity in particular still develop further into adolescence and young adulthood.

**Effects of sociocultural and task-specific aspects on children's narration.** Narration is not only an individual activity depending on cognitive and linguistic skills, but also a sociocultural one [55]. Multiple sociocultural factors, which affect children's familiarity with narratives and shape their understanding about how a narrative should be, might have influenced the findings of the present study.

One of these factors is culture. Cultures might differ in their narrative styles which determine which constituents of a narrative the narrators should focus on. Combined with the previous findings of Aksu-Koç and Tekdemir [1] and Küntay and Nakamura [10], the findings of the present study suggest that Turkish adult narrators focus more on the objective plot elements of the stories than their subjective evaluations although they were explicitly instructed to report the thoughts and emotions of the story character. This might be explained by Mesquita's [61] argument that the meaning of an emotional situation is regarded as obvious for every individual who is familiar with that situation in the Turkish culture. In their narratives, Turkish narrators might focus on the events in the plot structure and leave their evaluation at the implicit level, because they assume that the meaning of the events is obvious to their

listeners. The children might learn this plot-oriented narrative style through their socialization practices in story-telling and story-reading contexts. The quality of shared book reading interactions is also critical [62, 63] which might explain the inter-individual differences among children in our sample. It might be related to the socioeconomic status of the parents [64, 65]. Another important factor influencing children's familiarity with narratives is the education system [9, 66]. Thus, studying story-telling and reading practices in the different socioeconomic strata in Turkish culture and education system seems to be essential for a better understanding of Turkish-speaking children's narrative abilities.

Narration is also task-specific, because specific narrative tasks and contexts are claimed to require and encourage specific kinds of narrative and cognitive abilities as a result of which different developmental trajectories might be observed in different contexts [37, 67, 68]. In the present study, the participants told their narrative while looking at the pictures of a storybook. The interpretation of the present findings should be constrained to the particular narrative context.

## Limitations

The present study has several limitations. First of all, the assessment of executive function was limited. Due to time restrictions, only one task was used to assess executive function, namely inhibitory control. In addition to inhibition, shifting and updating were claimed to be necessary for narrative production [37]. Multiple tasks assessing these cognitive abilities might have offered a more detailed account of the relationship between executive function and narrative development. Furthermore, children's linguistic proficiency might be related to their narrative abilities. In the present study, only linguistic recursive ability was considered. However, other aspects of language such as receptive vocabulary [69] were not measured. The switch from the first-order ToM tasks in the youngest group of participants to the second-order ToM tasks in the older ones was another limitation. Because of the fact that these two types of tasks tap on different representational skills, the shift between them did not allow the assessment of developmental changes between 4 and 5 years of age which might be important in terms of socio-cognitive as well as narrative development. The last limitation was the unequal gender distribution in the age groups. There were more male than female participants in the youngest age group whereas there were more female than male participants in the seven- and eight-years-olds and the adult group.

## Conclusion

Taking a complexity approach, the present study with Turkish children provides insight into how the landscape of action and the landscape of consciousness are integrated in the development of narrative and which social, cognitive and linguistic factors support them. Only plot complexity matching with the landscape of action was found to develop significantly with age and to be modulated by gender. It was correlated with linguistic recursive ability and executive function indicating that construing the course of events is a resourceful process that is related to recursive abilities and cognitive monitoring including inhibition. Evaluative complexity corresponding to the landscape of consciousness was found to be stable between 4 and 11 years of age and not significantly related to any studied predictors. Syntactic complexity was predicted by executive function, but was not found to increase significantly in the studied developmental period. Although covering a broad age range from 4–11 years, the present study indicates that further developments in the domains syntactic and evaluative complexity are likely to occur in later school-age and adolescence until adulthood. Besides cognitive and (cross-) linguistic factors, taking a socio-cultural and gender perspective was found beneficial

—yet important questions remained unresolved. For instance, in recent years, studies supporting the role of ToM and executive function in the development of other narrative abilities such as narrative comprehension and narrative writing have been accumulating [38, 70]. Overall, the findings of the present study should be further extended in future studies comparing different narrative abilities in typologically different languages in different contexts for a more comprehensive understanding of narrative skills and their development.

## Supporting information

**S1 Fig. Hypothesized relations between cognitive abilities (predictors) and types of complexity (outcomes) during overall development from 4–11 years of age.**
(DOCX)

**S1 Table. Coding scheme for plot complexity.**
(DOCX)

**S2 Table. Categories of evaluative devices coded for evaluative complexity.**
(DOCX)

**S3 Table. Coding scheme for the subordinate clauses in narratives.**
(DOCX)

## Author Contributions

**Conceptualization:** Hale Ögel Balaban, Annette Hohenberger.

**Data curation:** Hale Ögel Balaban.

**Formal analysis:** Hale Ögel Balaban.

**Funding acquisition:** Annette Hohenberger.

**Investigation:** Hale Ögel Balaban.

**Methodology:** Hale Ögel Balaban, Annette Hohenberger.

**Supervision:** Annette Hohenberger.

**Visualization:** Hale Ögel Balaban.

**Writing – original draft:** Hale Ögel Balaban.

**Writing – review & editing:** Annette Hohenberger.

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
