## [Decision Letter · Decision Letter 0]

7 Nov 2019

PONE-D-19-23140

The development of narrative skills in Turkish-speaking children: A complexity approach

PLOS ONE

Dear Dr. Hohenberger,

Thank you for submitting your manuscript to PLOS ONE. After careful consideration, we feel that it has merit but does not fully meet PLOS ONE’s publication criteria as it currently stands. Therefore, we invite you to submit a revised version of the manuscript that addresses the points raised during the review process.

As you can see below, all three reviewers evaluated this work positively. They give excellent suggestions for improving the clarity and comprehensiveness of this paper. I would like to invite a revision that carefully considers and integrates the points raised by the reviewers. In particular, the reviewers raise important points with regard to presented analyses and procedures, including the presentation of underlying distributions of the data, the computation of inter-rater reliability and (in)dependent variables (including how they were coded), the presentation of statistical evidence (including statistical power, potential confounds [i.e. gender effects], and descriptive statistics), and the clarity of the experimental procedure. Addressing these points is essential to meeting PLOS ONE's publication criterion that all analyses are performed and described appropriately and rigorously. Reviewers 1 and 2 also provide excellent suggestions for better motivating and grounding the hypotheses with regard to previous findings. Reviewer 2 furthermore provides excellent suggestions for improving the clarity of the overall manuscript. Reviewer 3 raises important points (limitations) that should be discussed in the manuscript.

We would appreciate receiving your revised manuscript by Dec 22 2019 11:59PM. To enhance the reproducibility of your results, we recommend that if applicable you deposit your laboratory protocols in protocols.io, where a protocol can be assigned its own identifier (DOI) such that it can be cited independently in the future. For instructions see: http://journals.plos.org/plosone/s/submission-guidelines#loc-laboratory-protocols

We look forward to receiving your revised manuscript.

Kind regards,

Myrthe Faber

Academic Editor

PLOS ONE

Journal Requirements:

Reviewers' comments:

Reviewer's Responses to Questions

**Comments to the Author**

1. Is the manuscript technically sound, and do the data support the conclusions?

Reviewer #1: Partly

Reviewer #2: Yes

Reviewer #3: Yes

2. Has the statistical analysis been performed appropriately and rigorously? 

Reviewer #1: No

Reviewer #2: No

Reviewer #3: Yes

3. Have the authors made all data underlying the findings in their manuscript fully available?

Reviewer #1: Yes

Reviewer #2: Yes

Reviewer #3: Yes

4. Is the manuscript presented in an intelligible fashion and written in standard English?

Reviewer #1: Yes

Reviewer #2: Yes

Reviewer #3: Yes

5. Review Comments to the Author

Reviewer #1: Review: The Development of Narrative Skills in Turkish-speaking Children: A Complexity Approach

The research examines narrative complexity (evaluative, plot, and syntactic) in relation to theory of mind, linguistic recursive ability, and executive function (specifically, inhibitory control). Plot complexity showed significant development in early childhood and was predicted by linguistic recursive ability. The syntactic complexity of narratives was predicted by inhibitory control. The paper is well-written and it integrates timely literatures such as theory of mind and executive functioning with fundamental questions about narrative development.

The points that follow roughly correspond to the order that material appears in the paper.

1. The theoretical motivation—the rationale-- for the study is underdeveloped. Theory of mind, executive function, and linguistic recursive ability are individually expected to predict a particular component or components of narrative complexity (pg. 8). However, the reason (i.e., theoretical motive) for testing all three predictors for all three aspects of narrative complexity in one study is not clear (other than no one has previously brought all six factors together in a study). For instance, at the top of page 8, recursive abilities are briefly discussed without a clear rationale that ties those abilities to one or more of the specific narrative complexity factors. Next, however, at the end of the Introduction (line 181), recursive ability is predicted to impact syntactic complexity in narrative. Later, the impact of recursive ability on plot complexity is the focus of analysis (pg. 19) leading to a somewhat post hoc explanation of the link between the two variables (pgs. 22 and 23).

2. This is a minor point -- In the Introduction, it would be helpful to spell out more directly what ‘syntactic complexity’ means (i.e., are the authors talking about more than subordination?). For instance, reference is made (line 91) to developmental differences in syntactic complexity and it would be helpful to know the aspects they are referring to that bear on the present study.

3. As the authors note, the age groups are imbalanced in number and by gender. There were age by gender interactions (S1 File). Because of the relatively low number of 4-year-old participants (n = 18) and the necessity of administering a separate theory of mind task for that age group, the 4-year-olds’ results are analyzed separately from the other age groups (Table 2). The low statistical power limits the analyses (and conclusions) for this age group (lines 398-401, pg. 19).

4. Inter-rater reliabilities are reported for the theory of mind tasks and the executive function task. For the sake of clarity, it would be helpful to include a sentence spelling out that the rater reliability for these tasks refers to agreement on correct and incorrect responses (assuming this is the case). For the other variables (which involve more than pointing to an answer or forced-choice responding), it is typical to report Cohen’s kappa in addition to interrater reliability. There are a number of suggestions in the literature for what constitutes an acceptable kappa.

5. Please briefly clarify how the maximum score (19) is determined for the plot complexity coding (line 333, pg. 15).

6. The evaluative complexity categories do not include perception (S1 Table). In the Shiro research that the authors cite, reference to perception was the most common evaluative device used by children. The perception category seems especially relevant to the present study given the authors’ interest in the impact of theory of mind on evaluative complexity.

7. Please report the means and standard deviations by age group for the six major variables. It would also be helpful to see the zero order correlations among the variables. Without seeing those correlations, it is difficult to know whether there is a risk of multicollinearity in the regression analyses.

8. The S1 file contains some information (such as the age by gender interactions) that could be easily integrated into the main text.

9. Given the effect of gender on plot complexity (pg. 18), should gender be included as a predictor variable for the regression analysis for plot complexity (lines 402-403, pg. 19)? Also, if executive function was included in the regression analysis (line 402), it should be included in Table 3.

10. The Discussion could be more concise and focused on the specific contributions of the study.

Reviewer #2: The paper reports a study on how different types of narrative complexity relate to ToM, executive functioning and linguistic recursive ability. The study is well structured and well written, I would recommend somewhat minor revisions (with a few stronger points) to increase the clarity, accuracy and persuasive power of this paper.

1. Introduction

I was missing some arguments on the research gap this study would like to address, and the novel contribution to prior research. Later in the paper, there are some good points on this, I would suggest to relocate them here to make the introduction more convincing and less descriptive.

Also, at first read it was not clear that the “domains of complexity” in narratives refers to narratives generated by the participants, not ready made narratives. This should be explicit upfront to help the reader’s understanding.

2. Plot complexity

Based on the definition, it is not clear what makes this a complexity feature. In other words, what indicates higher and what indicates lower level of complexity. It only becomes clear towards the end of the paper.

I am missing a concluding sentence, how does this all relate to the current study. Are there any predictions the authors could make?

3. Evaluative complexity

Same as above. Based on the definition, it is not clear what makes this a complexity feature. In other words, what indicates higher and what indicates lower level of complexity. It only becomes clear towards the end of the paper.

Same as above. I am missing a concluding sentence, how does this all relate to the current study. The paragraph indicates you are doing comparison of nationalities. The authors should conclude this section with a prediction, or at least with a link to the current study.

4. Syntactic complexity

Line 88-87 implies that syntactic complexity is dependent of the other types of complexities. Theoretically it makes lots of sense, but it makes me wonder, if this affects the predictions. One would assume a high correlation among these variables.

Same as above. I am missing a concluding sentence, how does this all relate to the current study. The paragraph indicates you are doing comparison of nationalities. The authors should conclude this section with a prediction, or at least with a link to the current study.

5. Hypothesis

I am not convinced that all hypotheses are well grounded in the presented prior findings. I would suggest that the authors check and complete the argumentation whenever it is necessary.

There are many hypotheses, not always easy to follow, therefor it would be really helpful to see a visual overview of the conceptual model.

Line 177-181: this section breaks the flow of the hypotheses, relocate it to the method section.

6. Methods

Table 1, replace semi colons to dots when indicating mean age.

Checking gender distribution among age groups: it is mentioned in the limitations in one sentence that gender was not distributed equally among age groups. I would like to see the exact calculation here upfront. Also, I believe that gender should be always included as a covariate into all statistical tests to control for this confound. The authors mention that the effect of gender was not significant (line 356), which is not supported by data, so we don’t know to what extent it is not significant. Nevertheless, gender has to be included because of the confound.

Also, I would like to see information on the distribution of schools (the location of recruitment) among the age groups. It looks like a multilevel data, i.e. it can be that students learning in better schools have better mental skills. The school itself can have an influence on the dependent variables. The authors should reflect on this theoretically but perhaps also statistically.

The authors should provide a power analysis for their study. Non-significant effects might be due to power issues.

7. Materials

Why is section 207-211 called Materials? It belongs to the coding procedure discussed later.

Inter-rater reliability: throughout the paper, the authors are extremely vague about how inter-rater reliability was calculated. It seems they used simple percentage, which is not an adequate way to do it. I would suggest that authors recalculate IR reliability in terms of Krippendorf alpha or Cohen Kappa.

It is not clear at first how the sessions were recorded. Please, provide details.

8. Second order ToM task

One or two examples would help the reader understand how this measure works.

9. Comprehension of complement clauses task

When it comes to the result section it is difficult to trace back which construct was measured by which measure. I would suggest that to indicate the construct name in the sub-headings to make the readers’ task easier.

10. Syntactic complexity (line 345)

The authors say that the text was segmented into main and subordinate clauses. This can be tricky when verbal responses are transcribed and the transcriber puts there the punctuation that later serves as the basis for unit segmentation. Typically segmentation is also done by at least two people whose agreement is measured. I am very curious how the authors ensured that the segmentation is accurate (1. At the level of transcription 2. At the level of segmentation of the transcribed texts).

11. Results

The authors should be consistent about how they define marginally and non-significant values. Line 406 p = .054 is defined as marginally significant; Line 419 p =.06 is defined as non-significant.

12. Discussion

Line 512: the lack of statistical power --- what do the authors mean by this here, does not seem to fit the argument.

Line 536: Finnish is also an agglutinating language, it reads as if it was not and contrasted with Turkish.

Line 530-540: a few examples would help to follow the argumentation on the unique nature of Turkish language.

Around line 590: the argumentation on the plot oriented style. I would suggest that the authors also reflect on their methods at least in two ways:

1. How the instruction was formulated, I would say that it was formulated in a way that triggered plot-oriented narratives as opposed to evaluation-rich narratives.

2. Adults had to comment on a children book, which largely decreases the ecological validity of the narrative they generated. They most probably will not evaluate children book stories and characters not because of their cultural background but because of the simplicity of the story. Perhaps there are other studies of the same kind with Turkish adults that could help to understand what was going on.

13. Conclusion

As I mentioned above, gender is a confounding variable, which should be controlled for which might change the outcomes and hence the discussion.

Thank you for the opportunity of reviewing this paper.

Reviewer #3: This is a brilliant paper, and I am looking forward to seeing this published so that I can refer to it in my own work. I found it very enlightening and clearly structured. There are a few concerns that I would like to share with the authors in the hope that they can use my feedback in their final submission.

- I think it is important to pay attention to the possibility of intra-cultural differences (page 4, line 76) and where they might come from, either here or in the discussion. What would the parameters be of such differences? Exposure (quality/quantity) to narratives? Being read to as a child? Obviously narrative skills are essential, they seem to develop somehow, but… how? I very much hope the authors can share their knowledge about this. These parameters might also shed light on intercultural differences.

- It is important to communicate with the readers what the rationale is for choosing Turkish speaking children. Is there any particular expectation here about their development in comparison to others? Are there theoretical reasons to focus on them? It might be just practical reasons of course! And I do not mean to say the choice requires justification.

- Page 10, line 213, I fully understand that it is necessary to work with the same book for all groups. But having 10-11 year olds and adults respond to a picture book about a frog might pose a serious problem. Personal experience suggest that kids of that age are less than willing to be involved in picture books. Could the authors reflect on that?

Small issues:

- Page 2, line 44. I am not sure whether it is necessary to use caps. Also later on page 7. There are also other keywords that are important in this article, and it is unclear why these words are privileged with the use of caps.

- Page 8, line 177, maybe explain here briefly what an emotional Stroop Task is. Also, what “the rule” refers to might not be clear to all readers.

- Unclear to me what the numbers under M refer to in table 1. 4;4, should that not be 4.4? And M’s should be accompanied by SD’s.

- Page 9, line 196: “Most children” is a bit vague. Especially important considering the potential influence of background differences within a culture.

- Could the authors maybe provide a bit more information about the coding? One student did the coding? Where did the differences then occur? Between who and who? The student and the authors?

- Do mention somewhere the potential threats of validity when cells are so small (table 1: 2 adult males, only 7 females in age group 4).

- Maybe I overlooked the explanation of why adult participants had to be excluded from the analyses. (p. 16, line 352).

- Please check the quality of Figure 1. It was a bit hard to read when printed on paper.

6. PLOS authors have the option to publish the peer review history of their article (what does this mean?). If published, this will include your full peer review and any attached files.

Reviewer #1: No

Reviewer #2: Yes: Katalin Bálint

Reviewer #3: No

---

## [Author Response · Author response to Decision Letter 0]

10 Feb 2020

see our submitted file "Response_to_reviewers_09_02_2020"

---

## [Decision Letter · Decision Letter 1]

20 Apr 2020

The development of narrative skills in Turkish-speaking children: A complexity approach

PONE-D-19-23140R1

Dear Dr. Hohenberger,

We are pleased to inform you that your manuscript has been judged scientifically suitable for publication and will be formally accepted for publication once it complies with all outstanding technical requirements.

With kind regards,

Myrthe Faber

Academic Editor

PLOS ONE

Additional Editor Comments (optional):

Reviewers' comments:

Reviewer's Responses to Questions

**Comments to the Author**

1. If the authors have adequately addressed your comments raised in a previous round of review and you feel that this manuscript is now acceptable for publication, you may indicate that here to bypass the “Comments to the Author” section, enter your conflict of interest statement in the “Confidential to Editor” section, and submit your "Accept" recommendation.

Reviewer #2: All comments have been addressed

2. Is the manuscript technically sound, and do the data support the conclusions?

Reviewer #2: Yes

3. Has the statistical analysis been performed appropriately and rigorously? 

Reviewer #2: Yes

4. Have the authors made all data underlying the findings in their manuscript fully available?

Reviewer #2: No

5. Is the manuscript presented in an intelligible fashion and written in standard English?

Reviewer #2: Yes

6. Review Comments to the Author

Reviewer #2: The authors adequately addressed the major comments and answered all of my questions. I will be happy to see this study published.

7. PLOS authors have the option to publish the peer review history of their article (what does this mean?). If published, this will include your full peer review and any attached files.

Reviewer #2: Yes: Katalin E. Balint

---

## [Editor Report · Acceptance letter]

27 Apr 2020

PONE-D-19-23140R1 

The development of narrative skills in Turkish-speaking children: A complexity approach 

Dear Dr. Hohenberger:

I am pleased to inform you that your manuscript has been deemed suitable for publication in PLOS ONE. Congratulations! Your manuscript is now with our production department. 

With kind regards,

on behalf of

Dr. Myrthe Faber 

Academic Editor

PLOS ONE